# Occurrence of Antibiotic Resistant Bacteria in Flours and Different Plant Powders Used in Cuisine

**DOI:** 10.3390/foods11223582

**Published:** 2022-11-10

**Authors:** Júlia Koreneková, Monika Krahulcová, Klára Cverenkárová, Karol Červenčík, Lucia Bírošová

**Affiliations:** Department of Nutrition and Food Quality Assessment, Institute of Food Science and Nutrition, Faculty of Chemical and Food Technology, Slovak University of Technology, Radlinského 9, 81237 Bratislava, Slovakia

**Keywords:** flour, plant powder, dough, antibiotic resistance, biofilm

## Abstract

In recent years, several alimentary diseases have been connected with the consumption or tasting of raw flour and dough. Microbiological quality concern is also raising due to increased consumer demand for plant powders, while some of them can be consumed without prior thermal processing. In this study, we have focused on the occurrence of antibiotic-resistant coliform bacteria and enterococci in flour, plant powder and dough from Slovak retail. Our results indicated the presence of both total and antibiotic-resistant coliform bacteria and enterococci in the flour and powder samples. Lower numbers of the total, as well as resistant bacteria, were detected in flours compared to plant powders. Coliform bacteria isolates were predominantly identified as *Klebsiella* spp. and *Enterobacter* spp. Ampicillin resistance appeared in 97% of isolates followed by chloramphenicol resistance (22%) and tetracycline resistance (17%). The presence of the *bla*_SHV_ gene was confirmed in 13% of isolates. The *tetA* and *tetE* genes were present in 25% of isolates of coliform bacteria. The presence of enterococci was detected only in plant powders. Antibiotic-resistant strains were identified as the following: *Enterococcus casseliflavus*, *E. gallinarium* and *E. faecium*. Despite the isolates showing resistance to vancomycin, the presence of the *vanA* gene was not detected. The majority of antibiotic-resistant isolates belonged to the group of medium biofilm producers. None of these isolates showed efflux pump overproduction. Antibiotic-resistant coliform bacteria and enterococci were not detected in the processed doughs.

## 1. Introduction

Flour is classified as a food with low water activity, which ensures the growth inhibition of most microorganisms [1,2]. Nevertheless, it can be contaminated with pathogens, some of which are able to maintain viability long enough to cause disease [1,3]. For example, studies have shown that *Escherichia coli* and *Salmonella* sp. can survive 12 to 16 weeks in flour stored at room temperature [1]. The flour quality and safety are greatly affected by the initial microbiological quality of the cereal grains. A comparison of the final milling products of high and low microbiological quality wheat confirmed that flour with a higher number of microorganisms is obtained from lower microbiological quality wheat [4]. Conditioning is the process of adding water to wheat grains before milling to strengthen the bran and soften the endosperm. However, a long time (6–48 h), appropriate temperature (24–25 °C) and humidity during this process promote the growth and reproduction of the microorganisms’ present [5]. No significant increase in the total number of aerobic microorganisms was observed in the experimental system, but after conditioning, the presence of bacteria that were not previously detectable was detected, and a significant increase in *E. coli* was also demonstrated. This suggests selective enrichment with bacteria that were below the detection limit prior to conditioning or their recovery from a non-culturable but non-infectious state [6]. At the same time, there is a risk of cross-contamination by microorganisms potentially inhabiting the grinding equipment. Equipment used to purify and condition wheat has been shown to contain populations of aerobic mesophilic bacteria, *Enterobacterales*, and micromycetes [7]. As none of the operations in the flour recovery process involve chemical or thermal treatment to ensure safety, it can be expected that microorganisms originally present on cereal grains, including the possible presence of pathogenic microorganisms, will also be present in reduced quantities in ground cereal grains [4].

The Lower Saxon State Office for Consumer Protection and Food Safety carries out regular monitoring of the microbiological quality of flour. In 2017, shiga-toxigenic *E. coli* (STEC) was isolated from five retail flour samples [8]. As part of the Federal Monitoring Plan, the presence of STEC was confirmed in Germany in 2018 in 50 of the 328 samples of wheat, spelled and rye flour tested. A wide variety of STEC serotypes were detected in the samples, including human pathogenic types with different combinations of pathogenicity factors [6]. The presence of STEC was detected in 10.8% of 93 flour samples of different cereal varieties (wheat and mixtures of wheat with other cereal flour) purchased at retail markets in Switzerland [6,9].

In 2008, in New Zealand, 66 people suffered from salmonellosis due to consumption of an uncooked baking mixture containing flour contaminated with *S. Typhimurium* (STM42). This was the first time that contaminated flour had been identified by laboratory and epidemiological research as a source of *Salmonella* [10,11,12]. The first outbreak of flour-related STEC occurred in the United States in 2016. In 24 states, 63 patients were infected with STEC O121 or STEC O26 strains [3,13]. In 2017, there were 29 cases of the STEC disease in six provinces in Canada. This was the first nationwide *E. coli* epidemic associated with flour consumption in Canada [11,14]. These flour related epidemics point to the consequences of risky consumer behavior, in particular to the health risks associated with the consumption of raw products that are intended for heat treatment before consumption [12].

Lifestyle changes, together with the increasing incidence of celiac disease, have caused increased consumer demand for gluten-free products [15]. The market for gluten-free products is growing significantly as better diagnostic procedures make it possible to identify an increasing number of people with celiac disease and other disorders related to gluten sensitivity, such as dermatitis herpetiformis, gluten ataxia, wheat allergy and non-celiac gluten sensitivity [16]. In 2019, 22,735 patients with celiac disease were registered in Slovakia [17]. The basic treatment for these types of disorders is to follow a strict and permanent gluten-free diet. While patients suffering from gluten-related diseases rely on a gluten-free diet, other consumers choose it as a lifestyle because it is of cultural, environmental, civic, historical, ethnic or health concern. All these factors support the gluten-free market and its continued growth [16]. The current market provides several alternatives to flour (such as plant or insect powders) that can replace traditional wheat flour. Some of these flours can be consumed without previous heat treatment, for example, by adding to smoothie drinks, yogurts, cocktails, etc. [18]. However, if these products are contaminated with pathogenic microorganisms, they may pose a serious risk to human health.

According to these facts, we have decided to evaluate the microbiological quality of flours and powders as well as processed doughs available in Slovak retail. Since antibiotic resistance is a global emerging problem, we have focused on the detection and characterization of antibiotic-resistant coliform bacteria and enterococci in these foodstuffs used in common cuisine.

## 2. Materials and Methods

### 2.1. Sample Collection

In total, 60 samples of flours and powders, 29 of cereal flours and 31 of plant and insect powders were collected during 2019 and 2021 in Slovakia, predominantly Bratislava and subjected to microbial analyses. The abbreviation of each sample used in the paper and details of analyzed flours or powders are summarized in Table 1. Samples of 13 cooled doughs were also subjected to microbial analyses, which were obtained from markets in Bratislava, Slovakia. The abbreviation of each sample and details of analyzed doughs are summarized in Table 2. All tested samples were subjected to microbial analyses immediately after arriving at the laboratory. Samples of dough were retained at a cooling temperature of 8 °C during the time of transport to the laboratory.

### 2.2. Microbiological Analysis

Ninety mL of sterile peptone water solution (0.85% NaCl and 0.1% peptone) was homogenized with 10 g of sample of powder or flour or cooled dough [19,20,21]. Total aerobic bacteria, total and antibiotic-resistant coliform bacteria and enterococci were enumerated using the plate count agar (Biolife Italiana srl., Milan, Italy), Chromocult-coliform agar (VWR, Darmstadt, Germany) and Slanetz-Bartley agar (BioLife Italiana srl., Milan, Italy), respectively. Antibiotic-resistant colonies of coliform bacteria and enterococci were monitored by adding specific concentrations to the agar. Different concentrations were chosen according to resistance limits established by the European Committee on Antimicrobial Susceptibility Testing and in the case of antibiotic tetracycline according to Clinical and Laboratory Standard Institute [22,23]. To detect antibiotic-resistant coliform bacteria, antibiotics were used: ampicillin, gentamicin, chloramphenicol, ciprofloxacin, and tetracycline (Sigma-Aldrich, St. Louis, MO, USA). To detect antibiotic-resistant enterococci, antibiotics were used: ampicillin, gentamicin, ciprofloxacin, and vancomycin (Sigma-Aldrich, St. Louis, MO, USA). Agar plates with plate count agar were cultivated aerobically at 30 °C for 72 h. Chromocult coliform agars were cultivated aerobically at 37 °C for 24 h. Agar plates with Slanetz-Bartley agar were cultivated aerobically at 40 °C for 48 h. The results were expressed as the mean of all the repetitions together with the standard deviation (SD).

### 2.3. Identification of Antibiotic-Resistant Strains

The identification of antibiotic-resistant bacterial strains was performed by matrix-assisted laser desorption ionization-time of flight (MALDI-TOF) mass spectrometry (Bruker, Germany). After the monitoring of resistant coliform bacteria, colonies were randomly picked and isolated by the streak plate method on Mueller Hinton agar (Biolife, Italy) plates for 24 h at 37 °C (enterococci for 48 h at 40 °C). A sample of each strain (one colony) was applied on a steel target plate and covered with 1 μL of the matrix (saturated solution of α-cyano-4-hydroxycinnamic acid in 2.5% trifluoroacetic acid and 50% acetonitrile) and left to air-dry. The target plate was inserted into MALDI-TOF mass spectrometry, and analysis was performed via an AutoFlex I TOF-TOF apparatus (Bruker Daltonics Inc., Billerica, MA, US) in linear positive-ion mode (*m*/*z* range of 2000 to 20,000 with gating of ions below *m*/*z* 400 and a delayed extraction time of 450 ns). Gaining spectra were analyzed using MALDI BioTyper software (v 2.0) based on an algorithm for matching spectral patterns in logarithmic scores 0–3 (BioTyper Library v 3.0; Bruker Daltonics s.r.o., Brno, Czech Republic). A score above 1.9 ensured bacterial species identification by comparison of the obtained bacterial fingerprints with the existing database [24,25].

### 2.4. Susceptibility Testing of Resistant Isolates

The susceptibility of bacterial isolates to different antibiotics was detected with the macro-dilution drop method. Several ATBs of different classes were used for testing coliform bacteria (ampicillin, ceftazidime, ciprofloxacin, tetracycline, gentamicin, chloramphenicol, and meropenem). For enterococci, antibiotics ampicillin, gentamicin, ciprofloxacin, and vancomycin were used. The concentration of each ATB is listed in Table 3. Antibiotic concentrations were defined by resistant breakpoints according to EUCAST (R1) and according to CLSI (R2) [22,23]. The third concentration was selected to determine the highest level of resistance in coliform bacteria isolates as described previously by Krahulcová et al. [25]. The experiment was performed using Mueller-Hinton agar (Biolife, Italy). Incubation of plates was at 37 °C for 24 h for coliforms and 40 °C for 48 h for enterococci. The susceptibility profiles after incubation were evaluated visually [24,25]. Each experiment ran in triplicates and was repeated three times. For statistical analysis, the Student’s *t*-test was applied.

### 2.5. Biofilm Production Testing

Biofilm production assessment was performed according to Beenken et al. (2003) [26]. A sterile 96-well microtiter plate was inoculated with diluted overnight cultures of tested isolates (1:200 in tryptic soy broth). Incubation was static at 37 °C, 24 h for coliforms, and 40 °C, 48 h for enterococci. After incubation the overnight cultures were removed, and the wells of the microtiter plate were washed twice with 200 μL of PBS. The forming biofilm was fixed with 200 μL of 96% ethanol for few seconds and the microtiter plate was left to air-dry. Bacterial cells forming biofilm in the wells were stained with crystal violet solution (0.41% in 12% ethanol) for 3 min. After staining, wells were washed twice with PBS. As the final step, 200 µL of 96% ethanol was added to each well to dissolve the absorbed crystal violet in biofilm. The absorbance of crystal violet in each well was measured at 570 nm using a plate reader device (BioTek Inc., Seattle, WA, USA). Each experiment was repeated three times and ran in six parallels. For statistical analysis, the Student’s *t*-test was applied. The positive control strain was *Pseudomonas aeruginosa* (CCM 3955) considered strong producer of biofilm. Biofilm producers were then divided into 4 groups according to the values of measured absorbances: weak (<0.2), medium (0.2–0.3), strong (0.3–0.9), and very strong (>1.0) biofilm producers as described in Taniguchi et al. (2009) [24,25,27].

### 2.6. Efflux Pumps Overproduction Testing with Ethidium Bromide (EtBr)

The detection of efflux pump overproduction was evaluated via the EtBr-agar Carthweel method [28]. Detection was performed on Mueller-Hinton agar plates supplemented with ethidium bromide (2.5 mg/L). Inoculums in a density of 0.5 McFarland standard were prepared from overnight cultures and applied on marked EtBr plates with a swab. Incubation of inoculated EtBr-plates was at 37 °C for 16 h (coliforms) or 40 °C for 48 h (enterococci). The experiment was based on visual evaluation with UV irradiation due to fluorescence active compound EtBr (Sigma-Aldrich, St. Louis, MO, USA) [25,29]. The reference strain used as a comparative negative control for coliforms was *E. coli* (CCM 3988) and for enterococci *Staphylococcus epidermidis* (CCM 3953). Each experiment ran in triplicates and was repeated three times. For statistical analysis Student’s *t*-test was applied.

### 2.7. Detection of Antibiotic Resistance Genes

Resistance genes were detected via single and multiplex polymerase chain reaction (PCR). The presence of several resistance genes was detected in the bacterial isolates: β-lactamases TEM, SHV, OXA [30], tetracycline resistance genes Group II: *tetA*, *tetE* in coliforms [31] and vancomycin resistance gene *vanA* in enterococci [32]. The reaction mixture for multiplex PCR for detection of β-lactamases and *tet* genes consisted of 0,5 μL of each primer (0.25 μL for *tet* genes) and PCR water in a total volume of 25 μL. The reaction mixture for *vanA* PCR consisted of 0.25 μL of both primers, 10 μL of PCR buffer, 1 μL of deoxyribonucleotides mixture, 2 μL of 50 mmol MgCl_2_ solution and PCR water in total volume of 50 μL. A single colony was used as a DNA sample of tested isolates. Primers used during each PCR are listed in Table 4. PCR was carried out in a thermocycler (Mastercycler personal Eppendorf, Hamburg, Germany) using the following conditions for amplifying specific sections in β-lactamases: initial denaturation at 94 °C for 20 min; 30 cycles of denaturation at 94 °C for 40 s, annealing temperature 60.5°C for 1 min, 72 °C for 1 min and 30 s, and the final elongation step at 72 °C for 10 min. The same protocol was applied for the detection of *vanA* gene, except for an annealing temperature of 65 °C. The conditions used to determine tetracycline genes were initial denaturation at 94 °C for 20 min, 35 cycles of denaturation at 94 °C for 40 s, annealing temperature at 55 °C for 1 min, 72 °C for 1 min 30 s, and the final elongation step at 72 °C for 10 min. After initial denaturation, 25 μL of Multiplex PCR Master mix (Biotechrabbit, Berlin, Germany) for multiplex PCR or 0.5 μL of Hot Start Taq DNA Polymerase (Biotechrabbit, Berlin, Germany) for single PCR was added to the mixture [25,29].

PCR products were dyed with Loading Dye (Biotechrabbit, Berlin, Germany) and separated by gel electrophoresis (1.5% agarose gel in TAE buffer) set up at 100 V for 1 h and 40 min. Bands were visualized by subsequent staining in Gel Red (Biotium, Fremont, CA, USA) solution in TAE for 30 min and 100 bp DNA Ladder was used as a size marker. Positive controls used during the PCR reaction were subjected to sequence analyses to prove the presence of specific resistance genes.

## 3. Results and Discussion

### 3.1. Total Coliform Bacteria and Enterococci in Flour, Powder and Dough

Twenty-nine flour samples and 31 samples of plant or insect powders and 13 samples of dough were subjected to microbiological analysis. In 87% of samples of flour and powder were determined the total number of aerobic bacteria (Table 5). Fecal indicators (coliform bacteria or enterococci) were detected in 53% of samples of flour and powder. A total of 85% of samples of dough were determined total aerobic (Table 6). Coliform bacteria and enterococci were presented in 38% of all samples of dough, where total enterococci were observed only in one sample, namely sample of strudel dough in number of 4.1 log CFU/g.

The majority of detected numbers of total coliform bacteria (TCB) has ranged between 2.3 and 4.1 log CFU/g. These counts are comparable to Batool et al. (2012), who determined coliform bacteria counts in Pakistani flour in the range of 3.0–4.0 log CFU/g [33]. Twelve powder samples contained coliforms ranging from 2.3 to 5.3 log CFU/g. We recorded the most numerous growths in the case of chestnut powder (M3) originating from Italy. The microbiological quality of flour and other mill products is defined in the second part of the Codex Alimentarius of the Slovak Republic, which regulates the microbial requirements for food and packaging. This section sets out the requirements for coliform bacteria, fungi, and *Bacillus cereus*. As for coliform bacteria, these can be found in flours in the number of 103, the limit value being 104 in two samples out of five examined samples [34]. Microbiological criteria for powders are not currently available. Four samples of pizza and puff dough contained coliform bacteria (Table 6). In March 2022, the consumption of frozen pizza was connected to the *E. coli* outbreak in France [35]. *Escherichia coli* was detected only in one sample of pumpkin powder.

Compared to coliform bacteria, the number of samples with enterococci was much lower. We detected enterococci only in six samples of powder and one sample of strudel pastry. In cricket powder, we observed the highest count of enterococci. The genus *Enterococcus* spp. is part of the healthy intestinal microbiota of humans and animals, and some strains are used as starter cultures or probiotics. Due to their preferred intestinal habitat, their wide occurrence and ease of cultivation serve as indicators of fecal contamination and are part of hygienic standards for water and food products [36,37].

### 3.2. Antibiotic Resistant Coliform Bacteria and Enterococci in Flour, Powder and Dough

Besides TCB and TEC (total enterococci), we also detected their possible antibiotic resistance. When quantifying resistant coliform bacteria and enterococci, we applied the concentrations of individual antibiotics in accordance with EUCAST guidelines. In case of coliform bacteria, we applied the antibiotics ampicillin (penicillin), gentamicin (aminoglycoside), chloramphenicol (amphenicol), ciprofloxacin (fluoroquinolone), and tetracycline (tetracycline), which represent different classes of antibiotics. For tetracycline, we used the concentration specified by CLSI guidelines, while there are no breakpoints given by EUCAST. The antibiotics ampicillin (penicillin), gentamicin (aminoglycoside), ciprofloxacin (fluoroquinolone), and vancomycin (glycopeptide) were applied in the determination of resistant enterococci.

In 45% of flour samples, we detected antibiotic-resistant coliforms (Table 7). The majority was observed only in the case of ampicillin. Only in one sample of rye flour did we also observe gentamicin-resistant coliforms. In 39% of powder samples, we observed ampicillin resistance. We recorded the highest number in a sample of chestnut powder (M3). Significantly higher resistance to ampicillin was expected because most coliforms have intrinsic resistance to this antibiotic, except for the genus *Escherichia*. In five powder samples, we detected gentamicin resistance in the range of 2.7–4.3 log CFU/g. We recorded chloramphenicol and tetracycline resistance only in chestnut powder (M3). We did not detect resistance to ciprofloxacin in any sample. We have observed ampicillin-resistant enterococci only in six powder samples. In samples of chestnut (M3) and coconut powder (M19, M30), we also detected vancomycin-resistant enterococci. The only sample of insect flour-cricket flour (M49) contained ampicillin (penicillins) and gentamicin (aminoglycosides)-resistant coliform bacteria. Ampicillin-resistant enterococci were also present in this flour sample. In dough samples, we did not register any antibiotic-resistant bacteria.

The presence of total and antibiotic-resistant coliforms and enterococci in powders from chickpeas, rice, banana, almond, and coconut is of considerable concern, while these powders can be used in cold cuisine as an addition to drinks, spreads, salads, yogurt, smoothies, etc. [18].

In the United States in 2018, voluntary recalls of approximately 6300 packages of organic coconut powder were reported, due to the possible presence of *Salmonella*, which was detected in one package during routine testing. In the same year, *Salmonella* was the cause of the further withdrawal of flour from the market, namely 2099 packages of organic amaranth powder. Fortunately, none of these withdrawals have reported cases of disease due to flour consumption [38,39].

### 3.3. Identification of Antibiotic-Resistant Strains

Isolated colonies of antibiotic-resistant coliforms and enterococci were identified using a MALDI-TOF mass spectrometer, which can be used to analyze biological macromolecules by determining the weight of molecules after mixing with a matrix and laser beam ionization. Identification of microorganisms is performed by comparing the obtained mass spectrum with the spectra of reference strains in the relevant database. This method provides fast, well reproducible and accurate identification of microorganisms at the genus and species level [40].

In total, we identified 32 resistant coliform isolates belonging to six genera (Figure 1).

The majority was from the *Klebsiella* family (37%). Bacteria of the genus *Klebsiella* are widespread in nature. They are associated with several pathological processes in humans and animals but can also enter the food chain. *Klebsiella* is not considered to be a major intestinal or foodborne pathogen, but its participation in intestinal diseases cannot be ruled out. Multidrug-resistant *Klebsiella* spp. producing broad-spectrum β-lactamases that can be transported through food [41]. We identified two species of the genus *Klebsiella*, *K. oxytoca* and *K. pneumoniae*. These two species are the most common pathogens causing nosocomial infections in humans and are of great concern regarding the development of resistance to multiple antibiotics [42]. *Klebsiella pneumoniae* is a common cause of community-acquired and nosocomial infections, such as urinary tract infections, lower respiratory tract infections, and liver abscesses. Although not traditionally considered a foodborne pathogen, there have been reports of *K. pneumoniae* infections preceded by intestinal colonization, supporting the theory of food as a possible vector for the transmission of these pathogens [43]. All strains of *K. pneumoniae* were isolated from cricket powder.

Strains of *Cronobacter sakazakii* were isolated from chestnut and amaranth powder. *Cronobacter sakazakii* is an opportunistic pathogen associated with life-threatening neonatal infections that can lead to severe manifestations such as brain abscesses, meningitis, necrotizing enterocolitis, and systemic sepsis. Most cases are associated with the consumption of powdered infant formula contaminated with these microorganisms [44].

*Kosakonia cowanii* was isolated from teff and rye flour as well as from rice powder. *Kosakonua cowanii* is a recently reclassified bacterial species formerly known as *Enterobacter cowanii*. The bacterium is thought to be primarily a plant pathogen, but cases of human infections have been reported in which *K. cowanii* has been identified as the cause of rhabdomyolysis and bacteremia associated with rose thorns and as the cause of acute cholecystitis [45,46]. Up to 63% of *Enterobacter* were isolated from chestnut powder. These bacteria are ubiquitous and have been found in a wide variety of foods, including fruits and vegetables, meat, fish, eggs, tea, herbs, legumes, spices, dry animal feed, dairy products, powdered baby food, cereals, nuts, seeds, flour, pasta, chocolate, drinks, and water. *Enterobacter* cloacae is the most isolated species of the genus *Enterobacter* from humans and animals. It is referred to as an opportunistic pathogen identified in nosocomial infections [47]. *Escherichia vulneris* and *Siccibacter colletis* were isolated from chickpea powder. *Enterobacter vulneris* is an opportunistic human pathogen that has been identified in clinical cases of urosepsis, osteomyelitis, intravenous catheter-related bacteremia, meningitis, and dialysis-related peritonitis [48].

We have isolated and identified 10 antibiotic-resistant enterococci. The majority belonged to *E. casseliflavus* (70%), followed by *E.gallinarium* (20%) and *E. faecium* (10%) (Figure 2). *Enterococcus faecium* is the second most common species of *Enterococcus*, which is the cause of nosocomial diseases. It often possesses vancomycin resistance genes, making it a priority pathogen for the development of new antibiotics [49]. Most enterococci were isolated from coconut powder, only one isolate was derived from amaranth powder.

### 3.4. Characterization of Antibiotic-Resistant Strains

In almost all isolates we have detected ampicillin resistance (Table 8). Only one isolate of *K. cowani* from rye flour showed susceptibility to this antibiotic. The majority of coliform bacteria are intrinsically resistant to ampicillin, in the case of the genus *Escherichia* it is an acquired mechanism of resistance [24]. We did not observe resistance to gentamicin, ciprofloxacin and meropenem in any of the isolates. In seven isolates, we detected chloramphenicol resistance. In five *K. oxytoca* isolates from chestnut powder, we observed tetracycline resistance.

Enterococci have intrinsic and acquired resistance to most antibiotics used in humans, leading to treatment complications. Enterococci are naturally resistant to β-lactams, carbapenems and cephalosporins. They may also acquire resistance to tetracyclines, quinolones, macrolides, erythromycin, streptogramin, polymixins, clindamycin, and glycopeptides (vancomycin) [50]. Of the ten isolates tested, only three showed resistance to ampicillin at CLSI concentrations (Table 9). Ampicillin and penicillin, belonging to β-lactam ATBs, inhibit the synthesis of peptidoglycan, a critical component required for bacterial viability [50]. Higher levels of ampicillin resistance in enterococci are associated with the presence of the species-specific chromosomal gene pbp5, which encodes class B penicillin-binding proteins (PBPs) with binding affinity for ampicillin and cephalosporins [51]. All isolates were vancomycin-resistant. Vancomycin-resistant enterococci found in food can temporarily colonize the human gut and confer multidrug resistance to human *E. faecalis* via a conjugation plasmid [52].

Multidrug resistance, defined as resistance to three or more antimicrobials, was not detected in any of the isolates. The high level of antibiotic resistance is due to the widespread use of antibiotics in medical practice, especially in developing countries [42].

Efflux causes a decrease in intracellular drug concentrations, leading to subtoxic levels, which may ultimately induce phenotypic resistance [53]. All isolates of coliform bacteria and enterococci showed normal efflux. In contrast to our results, Krahulcova et al. recorded in their work the overproduction of efflux pumps in up to 19% of coliforms obtained from smoothie drinks [25].

About 1300 types of β-lactamases are currently known, of which 200 are types of broad-spectrum β-lactamases (ESBLs) [54]. ESBLs are commonly present in *Klebsiella* and *E. coli* species but have also been detected in other members of the *Enterobacterales* family, such as *Salmonella*, *Enterobacter*, *Citrobacter*, *Serratia* and *Proteus*. These microorganisms are the cause of septicemia, pneumonia, intra-abdominal abscess as well as various infections, e.g., urinary tract infections [55]. ESBL-related infections are on the rise and have been observed worldwide in hospitals and communities [56].

Coliform isolates, in which we confirmed ampicillin resistance were subjected to the detection of the *bla*_TEM_, *bla*_SHV_ and *bla*_OXA_ genes. We did not detect the presence of the *bla*_TEM_ and *bla*_OXA_ genes in any of the isolates. We confirmed the presence of the *bla*_SHV-1_ gene in 13% of ampicillin-resistant coliform bacteria, of which all isolates belonged to the *K. oxytoca* species (Table 10). *Klebsiella* species exhibit resistance to a broad spectrum of antibiotics, with resistance to penicillins, particularly ampicillin and carbenicillin, being mediated by the enzyme SHV-1 [57]. In the remaining 87% of isolates, ampicillin resistance may be due to other mechanisms or genes.

We also determined the presence of *tetA* and *tetE* genes in coliform isolates (Table 10). These genes mediate tetracycline resistance through membrane-associated proteins that export the drug from the cell and thereby reduce its intracellular concentration [58]. Although we did not detect excessive production of efflux pumps in any of the isolates, we recorded the presence of the *tetA* and *tetE* genes in eight isolates. It follows that the isolates have genes encoding specific efflux pumps, but they are not expressed. In addition to overproduction of efflux pumps, resistance to tetracycline antibiotics is mediated by two other mechanisms, namely the protection of ribosomes by proteins that prevent the efficient binding of tetracycline or the production of enzymes that inhibit the ATB. Each of these mechanisms is encoded by a wide variety of genes [58].

In the case of enterococci, we focused on the detection of the *vanA* gene, which is the most common gene encoding glycopeptide resistance [59]. Worldwide, the *vanA* gene is frequently detected in *E. faecium* and *E. faecalis* strains isolated from humans and animals but is less common in other species, such as *E. gallinarum*, *E. hirae*, and *E. durans* [60]. Although all isolates showed vancomycin resistance, we did not detect the *vanA* gene in any of the isolates. However, vancomycin resistance is mediated by several genes referred to as *vanA -B, -C, -D, -E, -G*. Of these, the *vanA* and *vanB* genes located on plasmids are most frequently detected, followed by chromosomally localized *vanD* and *vanC* genes. These genes encode changes in the amino acid sequence of the peptidoglycan precursor protein, thereby preventing vancomycin binding [61]. The relationship of food enterococci with clinical infections has not yet been clearly elucidated [62]. Foodborne enterococci are not known to be a direct source of resistant enterococci in humans but could pose a risk of transmitting resistance determinants to strains present in the human intestinal microbiota [60].

Biofilms can be found in food, medical and natural environments. Pathogenic bacteria that enter food production areas can remain there in the form of a biofilm covering the surfaces of machinery and equipment. Biofilms formed on these surfaces are a major cause of contamination of the final product, which ultimately leads to disease transmission, reduced food shelf life and quality, and thus huge economic losses [63]. Studies suggest that the biofilm is important for the transfer of conjugation plasmids due to the high proximity of cells in this structure. The resistance of a biofilm to ATB depends on various factors, such as physical, physiological, and genetic factors [63]. In a mature biofilm, bacterial cells can tolerate antibiotics at concentrations 10–1000-fold higher than those required to kill planktonic cells [64]. The majority of isolated coliforms (72%) were strong producers of biofilm (Figure 3), which correlates with the results of Krahulcová, who evaluated 73% of coliform isolates from raw milk and smoothie as strong biofilm producers [25]. The remaining 28% of isolates were medium-strength biofilm producers. We identified 86% of *K. oxytoca* as a strong biofilm producer. All *K. pneumoniae* isolates were classified as strong biofilm producers. Biofilm-forming *K. pneumoniae* is a major source of nosocomial infections associated with high morbidity and mortality due to limited treatment options [63]. Two isolates of *C. sakazakii* showed strong production, while the remaining three isolates were intermediate biofilm producers. The survival and persistence of *C. sakazakii* in powdered infant formula require the body’s ability to adapt to harsh osmotic and dry conditions. Studies have shown that *C. sakazakii* cells in biofilms were more persistent compared to plankton cells when exposed to low humidity conditions [65].

Enterococci are known for their ability to form biofilms [66]. Enterococcal biofilms have been reported in many infections, including urinary tract, wounds, gastrointestinal tract, and endocarditis. Biofilm-associated enterococcal infections are not only difficult to eradicate but also serve as a reservoir for antibiotic resistance genes. Low penetration of antibiotics through the biofilm matrix and the presence of persistent cells contribute to antibiotic tolerance of biofilms, leading to persistent infections [49]. Approximately 80% of persistent bacterial infections in the United States have been found to be associated with biofilms [63]. We observed medium biofilm production in isolates of *E. casseliflavus* and *E. gallinarium* (Figure 4). In the remaining 40% of isolates, including *E. faecium* isolate, we have detected strong production of biofilm. We did not observe very strong biofilm production in any of the coliform bacteria nor enterococci isolates (Figure 4).

## 4. Conclusions

The contamination of food with antibiotic-resistant pathogens poses a serious risk to public health. In recent years, flour has been recorded as an important source of food-borne diseases, while outbreaks were due to the consumption of raw dough. The new nutrition trends together with food supply globalization and increased civilizational disease incidence led to a change in consumers’ diet habits. For these reasons, we can observe an increased consumer demand for alternative flours or powders. Based on previous results, samples of flours and powders contain antibiotic-resistant bacteria where the majority were identified as *Klebsiella* spp. and *Enterobacter* spp. Although none of the resistant isolates harbored multidrug resistance and all showed normal efflux, we detected resistance genes such as *bla*_SHV_, *tetA* and *tetE*. Enterococci were less detected compared to coliform bacteria and were present only in non-cereal flours (powders). Identified strains were *E. casseliflavus*, *E. gallinarium* a *E. faecium*. Samples of dough contained only total coliform bacteria and enterococci. The presence of potentially pathogenic bacteria resistant to antibiotics in tested samples confirms the role of the food chain in the spread of resistance in the environment. Therefore, it is necessary to inform consumers about the correct handling of flour and dough and the importance of heat treatment. At the same time, we would like to suggest to legislatively define microbiological criteria for alternative plant-based flours, especially those which can be used without previous heat treatment.

## Figures and Tables

**Figure 1 foods-11-03582-f001:**
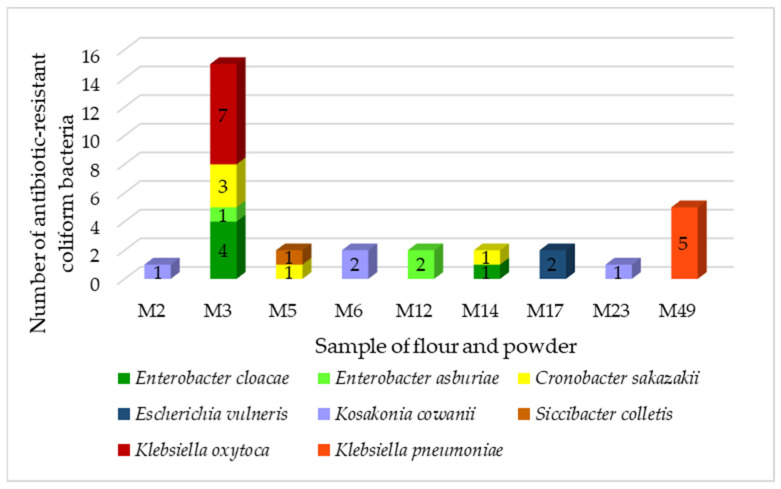
Antibiotic-resistant coliform bacteria isolated from flour, powder. M2—Teff, M3—Chestnut, M5—Amaranth, M6—Rice wholegrain, M12—Banana, M14—Chestnut, M17—Sorghum, M23—Rye, M49—Cricket.

**Figure 2 foods-11-03582-f002:**
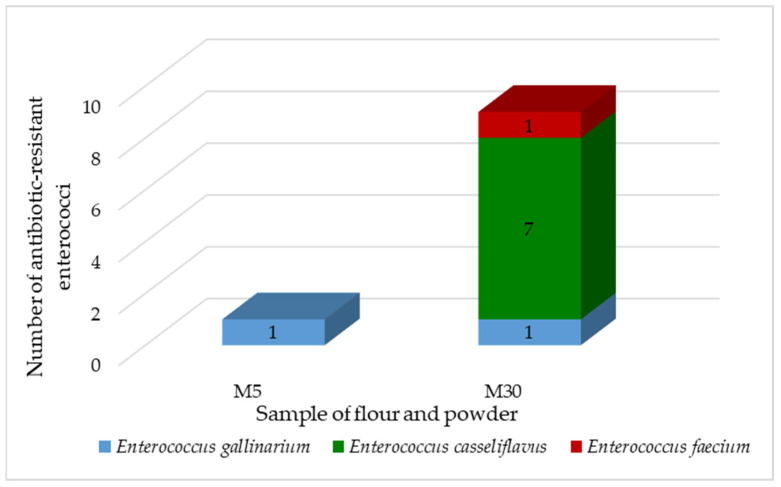
Antibiotic-resistant enterococci isolated from flour, powder. M5—Amarant, M30—Coconut.

**Figure 3 foods-11-03582-f003:**
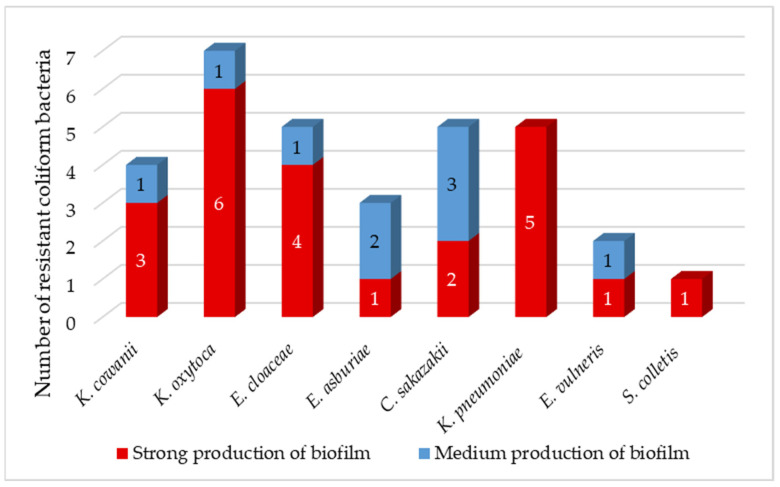
Production of biofilm between isolated resistant coliform bacteria.

**Figure 4 foods-11-03582-f004:**
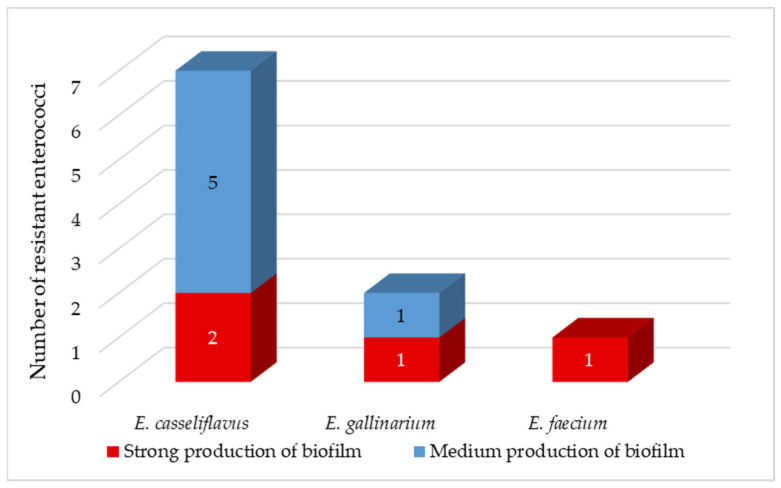
Production of biofilm between isolated resistant enterococci.

**Table 1 foods-11-03582-t001:** Details and types of flours and powders subjected for microbial analyses.

SP	Type of Flour or Powder	ED	SP	Type of Flour or Powder	ED
M1	Pumpkin	10.12.2021	M31	Smooth wheat	28.04.2022
M2	Teff	01.12.2021	M32	Poppy	20.02.2022
M3	Chestnut	30.11.2021	M33	Almond	31.07.2022
M4	Sorghum	07.06.2022	M34	Coarse wheat	12.08.2022
M5	Amaranth	30.12.2021	M35	Wholegrain spelt	03.05.2022
M6	Rice wholegrain	26.06.2022	M36	Wholegrain wheat	27.11.2021
M7	Almond	20.11.2021	M37	BIO wheat	09.06.2022
M8	Yellow pea	02.06.2022	M38	Smooth wheat	19.09.2022
M9	Green banana	17.05.2022	M39	Smooth wheat	31.03.2022
M10	Red lentils	02.06.2022	M40	Smooth wheat	25.02.2022
M11	Potato	28.05.2022	M41	Semi-coarse wheat	23.07.2022
M12	Banana	31.12.2022	M42	Smooth rye	22.05.2022
M13	Rice	26.11.2021	M43	Smooth wheat	04.07.2022
M14	Chestnut	28.02.2022	M44	Smooth wheat	17.03.2022
M15	Wheat	12.08.2022	M45	Smooth wheat	29.06.2022
M16	Buckwheat	17.11.2021	M46	Smooth spelt	22.01.2022
M17	Sorghum	09.03.2022	M47	Smooth wheat pizza	03.06.2022
M18	Almond	20.06.2022	M48	Wholegrain wheat	20.09.2022
M19	Coconut	31.08.2022	M49	Cricket	15.05.2021
M20	Oat	30.04.2022	M50	Flaxseed	05.12.2019
M21	Semi-coarse wheat	06.03.2022	M51	Pumpkin	08.12.2019
M22	Smooth wheat	15.03.2022	M52	Smooth spelt	23.04.2022
M23	Rye	29.03.2022	M53	Wholegrain spelt	08.03.2022
M24	Smooth wheat	30.03.2022	M54	Semi-coarse wheat	17.06.2022
M25	Hemp	11.05.2022	M55	Extra smooth wheat	16.08.2022
M26	Lupine	15.12.2022	M56	Smooth wheat	04.05.2021
M27	Apple	18.12.2023	M57	Semi-coarse wheat	16.08.2022
M28	Flaxseed	11.05.2022	M58	Smooth wheat	09.06.2022
M29	Pumpkin	30.01.2022	M59	Smooth wheat	06.05.2021
M30	Coconut	06.08.2022	M60	Semi-coarse wheat	13.08.2021

SP—sample of flour or powder; ED—expiration date; green color—information on the package allows consumer to use the product without previous heat treatment.

**Table 2 foods-11-03582-t002:** Details and types of dough subjected for microbial analyses.

SP	Type of Dough	ED	SP	Type of Dough	ED
C1	Strudel	06.11.2021	C8	Puff Pastry	15.11.2021
C2	Sourdough	06.11.2021	C9	Pizza	04.11.2021
C3	Puff Pastry	09.11.2021	C10	Pizza	14.11.2021
C4	Sourdough pizza	04.11.2021	C11	Puff Pastry	05.12.2019
C5	Puff Pastry	17.11.2021	C12	Pizza	05.12.2019
C6	Puff Pastry	17.11.2021	C13	Puff Pastry	17.11.2019
C7	Puff Pastry	03.11.2021			

SP—sample of dough; ED—expiration date.

**Table 3 foods-11-03582-t003:** Resistant breakpoints of ATBs used in susceptibility testing.

ATB Class	Antimicrobial	EUCAST (mg/L) >	CLSI (mg/L) ≥	Higher Than CLSI (mg/L)
Coliform bacteria				
Penicillins	Ampicillin	8	32	50
Cephalosporins	Ceftazidime	4	16	32
Carbapenems	Meropenem	8	4	12
Fluoroquinolones	Ciprofloxacin	0,5	1	2
Aminoglycosides	Gentamicin	2	16	20
Amphenicols	Chloramphenicol	8	32	50
Tetracyclines	Tetracycline	-	16	32
Enterococci				
Penicillins	Ampicillin	9	16	-
Fluoroquinolones	Ciprofloxacin	5	16	-
Aminoglycosides	Gentamicin	130	200	-
Glycopeptides	Vancomycin	5	16	-

ATB—antibiotic; EUCAST—The European Committee on Antimicrobial Susceptibility Testing; CLSI—Clinical Laboratory Standards Institute.

**Table 4 foods-11-03582-t004:** Resistant genes detected during PCR in resistant coliform isolates.

ATB Class	Gene	Primer	DNA Sequence 5′→3′	AS (bp)	AT (°C)
β-Lactams	*bla*_TEM_ *	fwd	CATTTCCGTGTCGCCCTTATTC	800	60.5
		rev	CGTTCATCCATAGTTGCCTGAC		
	*bla*_SHV_ *	fwd	AGCCGCTTGAGCAAATTAAAC	713	60.5
		rev	ATCCCGCAGATAAATCACCAC		
	*bla*_OXA_ *	fwd	GGCACCAGATTCAACTTTCAAG	564	60.5
		rev	GACCCCAAGTTTCCTGTAAGTG		
Tetracyclines	*tetA* **	fwd	GCTACATCCTGCTTGCCTTC	210	55
		rev	CATAGATCGCCGTGAAGAGG		
	*tetE* **	fwd	AAACCACATCCTCCATACGC	278	55
		rev	AAATAGGCCACAACCGTCAG		
Glycopeptides	*vanA* ***	EA1(+)	GGGAAAACGACAATTGC	732	65
		EA1(-)	GTACAATGCGGCCGTTA		

ATB—antibiotic; AS—amplicon size; AT—annealing temperature. * Dallene et al. 2018 [30], ** Ng et al. 2001 [31], *** Depardieu et al. 2004 [32].

**Table 5 foods-11-03582-t005:** Number of total aerobic and indicator bacteria in samples of flour and powder.

SP	TAB	TCB	TEC	SP	TAB	TCB	TEC
	log CFU/g				log CFU/g		
M1	<2.3	<2.3	<2.3	M31	4.2 ± 0.22	3.6 ± 0.19	<2.3
M2	<2.3	4.0 ± 0.19	<2.3	M32	3.5 ± 0.17	<2.3	<2.3
M3	<2.3	5.3 ± 0.25	3.8 ± 0.20	M33	<2.3	<2.3	<2.3
M4	3.6 ± 0.17	3.3 ± 0.14	<2.3	M34	2.6 ± 0.09	2.6 ± 0.13	<2.3
M5	4.6 ± 0.21	3.8 ± 0.15	2.9 ± 0.16	M35	2.9 ± 0.09	<2.3	<2.3
M6	4.8 ± 0.19	4.0 ± 0.22	ND	M36	3.4 ± 0.13	3.1 ± 0.17	<2.3
M7	2.7 ± 0.13	<2.3	2.4 ± 0.13	M37	4.2 ± 0.21	3.8 ± 0.10	<2.3
M8	3.5 ± 0.15	<2.3	<2.3	M38	3.5 ± 0.16	2.6 ± 0.13	<2.3
M9	3.9 ± 0.17	<2.3	<2.3	M39	3.4 ± 0.15	3.3 ± 0.16	<2.3
M10	4.1 ± 0.20	<2.3	<2.3	M40	3.4 ± 0.07	3.1 ± 0.14	<2.3
M11	3.2 ± 0.15	<2.3	<2.3	M41	2.7 ± 0.05	2.3 ± 0.08	<2.3
M12	4.5 ± 0.23	4.7 ± 0.15	<2.3	M42	4.1 ± 0.25	3.6 ± 0.16	<2.3
M13	4.1 ± 0.19	3.1 ± 0.17	<2.3	M43	3.2 ± 0.18	<2.3	<2.3
M14	3.8 ± 0.17	3.0 ± 0.09	<2.3	M44	3.7 ± 0.21	<2.3	<2.3
M15	3.3 ± 0.14	2.7 ± 0.07	<2.3	M45	2.8 ± 0.14	<2.3	<2.3
M16	4.8 ± 0.25	3.4 ± 0.11	<2.3	M46	2.7 ± 0.17	2.3 ± 0.06	<2.3
M17	4.6 ± 0.23	2.3 ± 0.06	<2.3	M47	2.7 ± 0.20	<2.3	<2.3
M18	3.9 ± 0.19	<2.3	<2.3	M48	2.3 ± 0.11	2.3 ± 0.05	<2.3
M19	4.0 ± 0.20	<2.3	3.3 ± 0.18	M49	5.2 ± 0.27	4.0 ± 0.15	4.7 ± 0.25
M20	4.1 ± 0.19	<2.3	<2.3	M50	3.0 ± 0.16	<2.3	<2.3
M21	3.7 ± 0.14	3.0 ± 0.12	<2.3	M51	2.7 ± 0.10	2.3 ± 0.04	<2.3
M22	4.2 ± 0.20	3.4 ± 0.15	<2.3	M52	3.1 ± 0.18	<2.3	<2.3
M23	4.4 ± 0.18	4.1 ± 0.23	<2.3	M53	<2.3	<2.3	<2.3
M24	4.2 ± 0.13	3.2 ± 0.13	<2.3	M54	<2.3	<2.3	<2.3
M25	2.3 ± 0.11	<2.3	<2.3	M55	2.4 ± 0.09	<2.3	<2.3
M26	3.1 ± 0.15	<2.3	<2.3	M56	2.4 ± 0.10	<2.3	<2.3
M27	3.4 ± 0.13	<2.3	<2.3	M57	3.0 ± 0.06	2.3 ± 0.02	<2.3
M28	3.5 ± 0.20	<2.3	<2.3	M58	<2.3	<2.3	<2.3
M29	2.7 ± 0.11	<2.3	<2.3	M59	3.0 ± 0.14	<2.3	<2.3
M30	4.0 ± 0.19	<2.3	3.5 ± 0.15	M60	<2.3	<2.3	<2.3

SP—sample of flour or powder, TAB—total aerobic bacteria, TCB—total coliform bacteria, TEC—total enterococci, green color—information on the package allows consumer to use the product without previous heat treatment.

**Table 6 foods-11-03582-t006:** Number of total aerobic bacteria, coliform bacteria, and enterococci in samples of dough.

SP	TAB	TCB	TEC	SP	TAB	TCB	TEC
	log CFU/g				log CFU/g		
C1	4.0 ± 0.22	<2.0	4.1 ± 0.22	C8	<2.0	<2.0	<2.0
C2	3.6 ± 0.18	<2.0	<2.0	C9	4.2 ± 0.24	2.6 ± 0.22	<2.0
C3	2.3 ± 0.09	<2.0	<2.0	C10	3.7 ± 0.19	2.3 ± 0.30	<2.0
C4	2.3 ± 0.14	<2.0	<2.0	C11	2.0 ± 0.11	<2.0	<2.0
C5	4.5 ± 0.26	<2.0	<2.0	C12	<2.0	<2.0	<2.0
C6	3.2 ± 0.16	2.6 ± 0.10	<2.0	C13	2.7 ± 0.09	<2.0	<2.0
C7	3.8 ± 0.21	2.6 ± 0.12	<2.0				

SP—sample of dough, TAB—total aerobic bacteria, TCB—total coliform bacteria, TEC—total enterococci.

**Table 7 foods-11-03582-t007:** Occurrence of antibiotic-resistant coliform bacteria and enterococci in samples of flour and powder.

SP	Antibiotic-Resistant Coliform Bacteria					SP	Antibiotic-Resistant Coliform Bacteria				
	AMP	CIP	CHF	GEN	TET		AMP	CIP	CHF	GEN	TET
	log CFU/g						log CFU/g				
M2	3.7 ± 0.15	<2.3	<2.3	<2.3	<2.3	M23	4.0 ± 0.21	<2.3	<2.3	3.3 ± 0.16	<2.3
M3	5.3 ± 0.31	<2.3	2.7 ± 0.09	4.3 ± 0.23	2.6 ± 0.11	M24	3.7 ± 0.18	<2.3	<2.3	<2.3	<2.3
M4	3.6 ± 0.24	<2.3	<2.3	<2.3	<2.3	M31	3.5 ± 0.20	<2.3	<2.3	<2.3	<2.3
M5	3.7 ± 0.22	<2.3	<2.3	2.7 ± 0.08	<2.3	M36	2.7 ± 0.09	<2.3	<2.3	<2.3	<2.3
M6	4.2 ± 0.28	<2.3	<2.3	3.2 ± 0.14	<2.3	M37	3.8 ± 0.12	<2.3	<2.3	<2.3	<2.3
M12	4.8 ± 0.27	<2.3	<2.3	3.6 ± 0.16	<2.3	M38	2.3 ± 0.10	<2.3	<2.3	<2.3	<2.3
M13	2.3 ± 0.11	<2.3	<2.3	<2.3	<2.3	M39	3.3 ± 0.14	<2.3	<2.3	<2.3	<2.3
M14	2.7 ± 0.14	<2.3	<2.3	<2.3	<2.3	M40	3.1 ± 0.17	<2.3	<2.3	<2.3	<2.3
M15	2.3 ± 0.09	<2.3	<2.3	<2.3	<2.3	M41	2.3 ± 0.11	<2.3	<2.3	<2.3	<2.3
M16	3.5 ± 0.17	<2.3	<2.3	<2.3	<2.3	M42	3.6 ± 0.18	<2.3	<2.3	<2.3	<2.3
M17	2.9 ± 0.22	<2.3	<2.3	<2.3	<2.3	M48	2.3 ± 0.13	<2.3	<2.3	<2.3	<2.3
M21	3.5 ± 0.24	<2.3	<2.3	<2.3	<2.3	M49	2.3 ± 0.16	<2.3	<2.3	3.6 ± 0.11	<2.3
M22	3.5 ± 0.22	<2.3	<2.3	<2.3	<2.3						
**SP**	**Antibiotic-Resistant Enterococci**										
	**AMP**		**CIP**		**GEN**			**VAN**			
	**log CFU/g**										
M3	2.0 ± 0.05		<2.0		<2.0			2.4 ± 0.18			
M5	2.0 ± 0.30		<2.0		<2.0			<2.0			
M7	2.0 ± 0.03		<2.0		<2.0			<2.0			
M19	2.3 ± 0.09		<2.0		<2.0			2.3 ± 0.15			
M30	3.5 ± 0.17		<2.0		<2.0			2.7 ± 0.20			
M49	4.5 ± 0.22		<2.0		<2.0			<2.0			

SP—sample of flour or powder, AMP—ampicillin, CIP—ciprofloxacin, CHF—chloramphenicol, GEN—gentamicin, TET—tetracycline, VAN—vancomycin, green color- information on the package allows consumer to use the product without previous heat treatment.

**Table 8 foods-11-03582-t008:** Antibiotic-resistant profiles of isolated resistant coliform bacteria against selected types of antibiotics.

SP	Resistant Isolate	Tested Antibiotics							SP	Resistant Isolate	Tested Antibiotics						
		A	G	C	CH	T	CF	MR			A	G	C	CH	T	CF	MR
M2	*K. cowanii*	R	S	S	S	S	S	S	M5	*C. sakazakii*	R	S	S	S	S	S	S
M3	*E. cloaceae*	R	S	S	S	S	S	S	M5	*S. colletis*	R	S	S	S	S	S	S
M3	*K. oxytoca*	R	S	S	S	S	S	S	M6	*K. cowanii*	R	S	S	S	S	S	S
M3	*K. oxytoca*	R	S	S	R	R2	S	S	M6	*K. cowanii*	R	S	S	S	S	S	S
M3	*K. oxytoca*	R	S	S	R	R2	S	S	M12	*E. asburiae*	R	S	S	S	S	S	S
M3	*K. oxytoca*	R	S	S	R	R2	S	S	M12	*E. asburiae*	R	S	S	S	S	S	S
M3	*K. oxytoca*	R	S	S	R	R2	S	S	M14	*E. cloaceae*	R	S	S	S	S	S	S
M3	*K. oxytoca*	R	S	S	R	R2	S	S	M14	*C. sakazakii*	R	S	S	S	S	S	S
M3	*C. sakazakii*	R	S	S	S	S	S	S	M17	*E. vulneris*	R	S	S	S	S	S	S
M3	*C. sakazakii*	R	S	S	S	S	S	S	M17	*E. vulneris*	R	S	S	S	S	S	S
M3	*C. sakazakii*	R	S	S	R	S	S	S	M23	*K. cowanii*	S	S	S	S	S	S	S
M3	*K. oxytoca*	R	S	S	R2	S	S	S	M49	*K. pneumoniae*	R	S	S	S	S	S	S
M3	*E. cloaceae*	R	S	S	S	S	S	S	M49	*K. pneumoniae*	R	S	S	S	S	S	S
M3	*E. asburiae*	R	S	S	S	S	S	S	M49	*K. pneumoniae*	R	S	S	S	S	S	S
M3	*E. cloaceae*	R	S	S	S	S	S	S	M49	*K. pneumoniae*	R	S	S	S	S	S	S
M3	*E. cloaceae*	R	S	S	S	S	S	S	M49	*K. pneumoniae*	R	S	S	S	S	S	S

SP—sample of flour and powder, A—ampicillin, G—gentamicin, C—ciprofloxacin, CH—chloramphenicol, T—tetracycline, CF—ceftazidime, MR—meropenem.

**Table 9 foods-11-03582-t009:** Antibiotic-resistant profiles of isolated resistant enterococci against selected types of antibiotics.

SP	Resistant Isolate	AMP	GEN	CIP	VAN
M5	*E. gallinarium*	S	S	S	R1
M30	*E. casseliflavus*	R2	S	S	R2
M30	*E. casseliflavus*	R2	S	S	R2
M30	*E. faecium*	R2	S	S	R2
M30	*E. casseliflavus*	S	S	S	R1
M30	*E. casseliflavus*	S	S	S	R1
M30	*E. gallinarium*	S	S	S	R1
M30	*E. casseliflavus*	S	S	S	R1
M30	*E. casseliflavus*	S	S	S	R1
M30	*E. casseliflavus*	S	S	S	R1

SP—sample, AMP—ampicillin, GEN—gentamicin, CIP—ciprofloxacin, VAN—vancomycin.

**Table 10 foods-11-03582-t010:** Detected resistance genes in resistant coliform bacteria.

Isolate Positive for Resistance Genes	SP	β-Lactamase Genes	*Tet* Genes	
		*bla* _SHV_	*tetA*	*tetE*
*K. oxytoca* (n = 6)	M3	1	5	
*E. asburiae* (n = 2)	M12			2
*E. vulneris* (n = 1)	M17			1
*K. pneumoniae* (n = 3)	M49	3		

## Data Availability

Data of the current study are available from the corresponding author. The data are not publicly available due to privacy.

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
