# Peer review of "Occurrence of Antibiotic Resistant Bacteria in Flours and Different Plant Powders Used in Cuisine"

_foods, 2022, doi:10.3390/foods11223582_

Round 1
Reviewer 1 Report
In the results, I also recommend the presentation of antibiotics by class of antibiotics. It is interesting to know which is the predominant group of antibiotics in insect flours, flours that are the subject of many researches in the field of human nutrition.Author Response
Dear Reviewer 1,
Thank you very much for your advice, we have completed classes of used antibiotics in the results section. Also, we have added most prevalent antibiotic resistance in insect flour, as the subject of many human nutrition studies.
Best regards, Krahulcová et al.

Reviewer 2 Report
In the manuscript by Koreneková et al. authors collected commercial samples of flours and powders from different plants and insects to evaluate whether they contain occurrence of antibiotic resistant bacteria. They draw an extensive classical microbiological analysis, testing several antibiotics in agar plates and identifying resistant isolates with MALDI and Bruker BioTyper Library. Resistant isolates were characterized for biofilm production, efflux pumps overproduction and eventually resistance genes were screened via multiplex PCR to elucidate the responsible gens.
The authors found that resistant bacteria were present in a very large fraction of flours and powders but not in processed doughs. They found a higher proportion of coliforms rather than enterococci and identified 8 major antibiotic resistant species. The authors conclude by stating that food labeling should cover such aspects and warn against the correct handling of flour and dough and the importance of heat treatment.
The manuscript is well written, clear in the scopes and adequately introduced. The methods are clear, exhaustive and rational and cover all aspects of the presented results. The results appear consistent with the methods and the figures/tables proposed. The references are well covering all aspects of the overall message and methods.
The choice of merging results and discussion seems appropriate give then overall message.
The quality, clarity and resolution of figure and their caption is adequate (although, personally, I'm not a big fan od 3D barplots, but this is just my opinion).
Author Response
Dear Reviewer 2,
Thank you very much for your opinion.
Best regards, Krahulcová et al.

Reviewer 3 Report
the purpose of this work was to identify the occurrence of antibiotic resistant colifom bacteria and enterococci derived isolated from sixty samples of flour and powders collected in two consecutive years in retail outlets in Slovakia. This work is a field survey and does not contain any experimental design; however, it provides very interesting information about the microbiological quality of flours and powders in retail outlets, and could thus be used to identify potential foodborne outbreaks. In my opinion, the work is straightforward and the manuscript is well-written and easy to follow. The conclusions of the authors are supported by the research data obtained. I have a minor comment to make before finalizing this manuscript. In Tables 6 and 7, the term "ND" must be replaced by the detection limit of the plating method. For example in table 6 for TCB, ND should be replaced by <XX log CFU/g depending on the threshold of the plating method and the technique (spread or pure) employed for the microbiological analysis.
Author Response
Dear Reviewer 3,
Thank you very much for your advice. We have replaced the term ND (not detected) in tables according to technique and plating method.
Best regards, Krahulcová et al.
